# GeneBench: Systematic Evaluation of Genomic Foundation Models and Beyond

## Abstract

The Genomic Foundation Model (GFM) paradigm is expected to facilitate the extraction of generalizable representations from massive genomic data, thereby enabling their application across a spectrum of downstream applications. Despite advancements, a lack of evaluation framework makes it difficult to ensure equitable assessment due to experimental settings, model intricacy, benchmark datasets, and reproducibility challenges. In the absence of standardization, comparative analyses risk becoming biased and unreliable. To surmount this impasse, we introduce GeneBench, a comprehensive benchmarking suite specifically tailored for evaluating the efficacy of Genomic Foundation Models. GeneBench offers a modular and expandable framework that encapsulates a variety of state-of-the-art methodologies. Through systematic evaluations of datasets spanning diverse biological domains with a particular emphasis on both short-range and long-range genomic tasks, firstly including the three most important DNA tasks covering Coding Region, Non-Coding Region, Genome Structure, etc. Our results on GeneBench have led to an interesting discovery: regardless of the number of parameters, the noticeable variation in preference between attention-based and convolution-based models for short—and long-range tasks could offer valuable insights for the future development of GFM. As a result, we propose a straightforward modified model called Genhybrid, which is an effective and efficient SSM-attention hybrid model suitable for all tasks we covered.

## 1 Introduction

Recently, significant advancements have been made in genomic research by utilizing foundation models (FMs) to analyze unstructured whole genome data. These genomic foundation models play a crucial role in various tasks such as predicting gene locations and functions, identifying regulatory elements, and studying species evolution (Ji et al., 2021; Fishman et al., 2023; Zvyagin et al., 2023).

Despite the importance of modeling Genomics foundations and the advancement of different training methods, there is still a noticeable absence of a thorough benchmark in this area that encompasses a range of practical application scenarios and different foundational model structures. The current benchmark either restricts its scope to short distances or oversimplifies the challenge by focusing solely on the classification task (Ji et al., 2021; Marin et al., 2023; Fishman et al., 2023). Moreover, with the influx of long sequence models called state space models (SSMs) (Gu & Dao, 2023; Liu et al., 2024b; Nguyen et al., 2024; Schiff et al., 2024), a systematic approach to evaluating up-to-date GFMs and inspiring subsequent development is also sorely lacking. Based on the current state of research, we thus summarise the three key problems in current GFMs: **(1) Incomplete evaluation:** Long sequence processing is crucial for modeling genetic data. Current tests of these models for long-sequence gene tasks are incomplete. **(2) Chaotic training strategies:** The variety of tokenizations and pre-training methods lack a fair platform to compare and select the most appropriate training method for DNA data. **(3) Snowy model design:** Do the various attention-based and recent state-space/convolution models have unique strengths in analyzing DNA? We need an inspiring experience that will influence future designs.

To address these issues, we propose GeneBench, a comprehensive benchmarking suite covering the three main genomic directions from short to long, as shown in Figure 1: the coding regions, the non-coding regions, and the genome architecture. GeneBench aims to establish a standardized platform

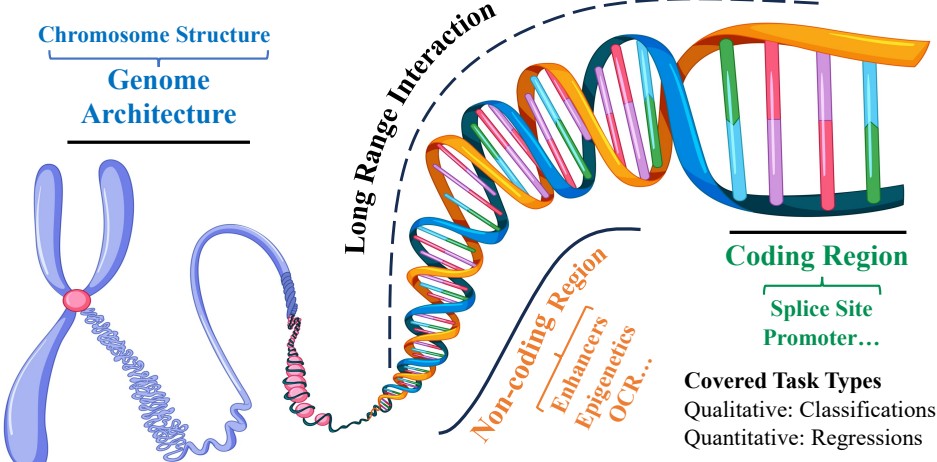

Figure 1: The illustration of the GeneBench covers the most representative tasks in eukaryotic genomic DNA, highlighting three critical components from short-range to long-range dependencies: the Coding Region, the Non-coding Region, and Genome Architecture. Notably, long-range interactions in DNA are very important in life processes.

for assessing the capability of Genomic Foundation Models (GFMs) to represent complex genome data accurately and to promote the development of this emerging field. Specifically, GeneBench includes ten widely-used GFMs and conducts comprehensive experiments using forty-four real-world datasets counted in Appendix D. Furthermore, with the guidance of our experimental findings, we propose a simple yet efficient hybrid model design that enjoys the two characteristics of performance in quadratic attention and efficiency in linear variants. Our contribution is summarized as follows:

- **Comprehensive Forty-five datasets:** We first integrate short- and long-range tasks covering three main aspects of genomics: non-coding region, coding region, and genome architecture.

- **Experiments covering various types of GFMs:** Investigate the impact of employing attention and SSM/convolution models in genomic modeling on different scales.

- **A modular and expandable code framework:** Provide a unified experimental environment to achieve fair comparisons and facilitate subsequent development of new methods.

- **Experimental results and new model:** Independent of the number of parameters, GFMs based on SSM/convolution and attention structures have their strengths in downstream tasks with different features. Therefore, we propose a simple yet efficient hybrid model that enjoys two worlds.

## 2 BACKGROUND AND RELATED WORK

### 2.1 PROBLEM DEFINITION

**Pre-training** We present the formal definition of the genomic modeling problem as outlined below. During the pretraining phase, the input $\mathcal{X}^{l,L} = \left\{ \boldsymbol{x}^i \right\}_{l-L+1}^{l} \in \{A, G, C, T, N\}$ up to position $l$ covers the past $L$ frames of base pairs, comprising Adenine (A), thymine (T), cytosine (C), guanine (G), or not known (N) nuclotides. At this stage, specific portions of nucleotides are masked for prediction purposes, either predicting the masked nucleotides or the subsequent nucleotide $\left\{ \boldsymbol{x}^i \right\}_{l'-L'+1}^{l'}$ from position $l'$, formulated as Eq. 1. The DNA sequence is initially encoded by a tokenizer into tokenization tensors $\mathcal{Z}^{l,L,D} \in \mathbb{Z}^{L \times D}$ with a hidden dimension of $D$.

$$\mathcal{L}_{\text{pretrain}} = -\sum_{j=l'-L'+1}^{l'} \log P(\boldsymbol{x}^j \mid \mathcal{X}^{l,L}) \qquad (1)$$

Table 1: Classification of the supported Genomic foundation models in GeneBench. The use of BERT in training strategy refers to training the model to predict the masked token in a sequence, while employing GPT in training strategy involves utilizing the next token prediction. Additionally, we have included expert models for particular downstream tasks that utilize One-hot encoding and training from the beginning for comparison.

| Category | Model | Conference/Journal | Tokenizer | Training strategy |
|---|---|---|---|---|
| Attention-based | DNABERT | Bioinformatics 2021 | K-mer | BERT |
| | Nucleotide Transformer | BioRxiv 2023 | K-mer | BERT |
| | DNABERT2 | ICLR 2024 | BPE | BERT |
| | GENA-LM | BioRxiv 2023 | BPE | BERT |
| SSM/Convolution-based | Hyena-DNA | NeurIPS 2024 | Char | GPT |
| | Caduceus | ICML 2024 | Char | BERT |
| | CNN | BMC Genomic Data 2023 | One-hot | Scratch |
| | SpliceAI | Cell 2019 | One-hot | Scratch |
| | DeepSTARR | Nature genetics 2022 | One-hot | Scratch |
| | Orca | Nature genetics 2022 | One-hot | Scratch |
| Hybrid | GenHybrid | Ours | K-mer | BERT |

**Fine-tuning** The model with learnable parameters $\Theta$ establishes a mapping $\mathcal{F}_\Theta : \mathcal{Z}^{l,L,D} \mapsto \mathcal{Y}$ by ultilizing nucleotide dependencies. In this context, $\mathcal{F}_\Theta$ represents a neural network trained to minimize the difference between the predicted target and the pretrained model. The optimal parameters $\Theta^*$ in one specific downstream task are determined as:

$$\Theta^* = \arg\min_\Theta \mathcal{L}\left(\mathcal{F}_\Theta\left(\mathcal{Z}^{l,L,D}\right), \mathcal{Y}\right), \tag{2}$$

where $\mathcal{L}$ is a loss function that measures this disparity. In this research, we classify prevalent downstream tasks into two categories: classification and regression. For the classification task, the target prediction is the discrete representation of the input genomic sequence $\mathcal{Y} \in \{0,1\}^C$, where $C$ represents the number of classes. In regression tasks, target prediction is the numerical tensor $\mathcal{Y} \in \mathbb{R}$.

## 2.2 GENOMIC FOUNDATION MODELS

**Attention-based** Foundation models in deep learning are trained on extensive data sets using self-supervised learning. The importance of these models has grown due to their capacity to leverage large amounts of unlabeled data. For instance, DNABERT by Ji et al. (2021) was developed based on the BERT model (Devlin et al., 2018) with a k-mer genomic tokenizer. Additionally, Benegas et al. (2023) introduced the Genomic Pre-trained Network (GPN) for predicting non-coding variant effects, surpassing supervised learning methods. Researchers have explored different approaches to enhance performance. For example, Dalla-Torre et al. (2023) introduced NT (Nucleotide Transformer), a genomic model with billions of parameters. On the other hand, researchers focus on optimizing model components and efficiency. DNABERT-2 (Zhou et al., 2023) replaces k-mer tokenization with Byte Pair Encoding (BPE) (Sennrich et al., 2015).

**SSM/Convolution-based** Despite the computational cost associated with scaling up in sequence length due to the quadratic complexity of attention mechanisms, there is room for more efficient alternatives. HyenaDNA (Nguyen et al., 2024) and Caduceus utilize the hyena operator (Poli et al., 2023) and state space model (Gu & Dao, 2023; Liu et al., 2024a) with a complexity of $\mathcal{O}(L\log_2 L)$ and $\mathcal{O}(L)$, significantly lower than the $\mathcal{O}(L^2)$ of attention-based models. The mathematical form of the above basic modules is listed in Appendix B.2.

## 3 BENCHMARKS AND METHOD

### 3.1 OVERVIEW

GeneBench has benchmarked eleven key genomic foundational models within a cohesive framework, comprising four attention-based models, six convolution-based models, and one hybrid model we designed. These models are outlined in Table 1, which details the associated conference/journal, the tokenizer types used, and their respective training strategies. The initial attention-based model employs a k-mer tokenizer, whereas the more recent attention-based model utilizes Byte Pair Encoding (BPE). HyenaDNA and Caduceus adopt a Char tokenizer due to their subquadratic space com-

Table 2: The dataset statistics for the tasks facilitated by GeneBench are meticulously detailed, delineating the various types of analyses supported. Within the typological column, the term "Sequence Binary Classification" refers to the assignment of an entire input sequence to one of two exclusive categories, thereby yielding a dichotomous classification outcome. In contrast, "Sequence Multi-class Classification" encompasses a more expansive classification, where an input sequence is allocated to one among a plurality of classes, surpassing the binary distinction. Furthermore, "Token Multi-class Classification" signifies a classification that operates at the token level, providing a nuanced categorization with multiple potential outcomes for individual elements within the sequence. Lastly, "Regression" denotes predicting a continuous range of values, as opposed to classes.

| Benchmark | Tasks | Type | Training size | Testing size | Length |
|---|---|---|---|---|---|
| Short Range | Mouse Enhancers | Binary classification | 968 | 242 | 500 |
| | Coding vs Intergenomic | Binary classification | 75K | 25K | 500 |
| | Human vs Worm | Binary classification | 75K | 25K | 500 |
| | Human Enhancers Cohn | Binary classification | 20K | 7K | 500 |
| | Human Enhancers Ensembl | Binary classification | 123K | 3K | 500 |
| | Human Ensembl Regulatory | Multi-class classification | 231K | 57K | 500 |
| | Human Nontata promoters | Binary classification | 27K | 9K | 500 |
| | Human OCR Ensembl | Binary classification | 14K | 35K | 500 |
| | Drosophila Enhancers Prediction | Regression | 402K | 41K | 128 |
| | Human Core Promoter Detection | Binary classification | 95K | 12K | 70 |
| | Human Transcription Factor Prediction | Binary classification | 128K | 5K | 30 |
| | Human Promoter Detection | Binary classification | 95K | 12K | 70 |
| | Human Splice Site Detection | Multi-class classification | 36K | 5K | 80 |
| | Mouse Transcription Factor Prediction | Binary classification | 80K | 10K | 30 |
| | Yeast Epigenetic Marks Prediction | Binary classification | 230K | 29K | 128 |
| | Virus Covid Variant Classification | Multi-class classification | 73K | 9K | 256 |
| | Central Dogma | Binary classification | 16K | 4K | 400 |
| | Mutation on Coding DNA | Regression | - | 10K | 1500 |
| Long Range | Splice Site Prediction | Multi-class classification | 146K | 16K | 15K |
| | Species Classification | Multi-class classification | 1K | 500 | 80M |
| | Promoters Prediction | Binary classification | 41K | 12K | 8K |
| | Genomic Structure Prediction | Regression | 21 | 3 | 256M |
| | Bulk RNA Prediction | Regression | 23K | 990 | 196K |

plexity and linear space complexity. Besides, we have included expert models for particular downstream tasks for comparison, named SpliceAI (Jaganathan et al., 2019), DeepSTARR (de Almeida et al., 2022), CNN (Grešová et al., 2023), and Orca (Zhou, 2022).

This design closely resembles the conventional deep-learning-based language models Devlin et al. (2018); Brown et al. (2020), but with modifications to the tokenizers tailored for genomic sequences, taking into account the simpler structure of genomes compared to human language. In general, the tokenizer converts a sequence of nucleotides into tokens. Each token is then converted into a numerical vector and represented as a matrix $M$ through embedding. Depending on the method used to segment the nucleotide sequences, tokenizers can be divided into k-mer tokenizers and BPE tokenizers. A newer approach has also emerged, where each individual nucleotide is directly mapped, known as the 'char tokenizer.'

## 3.2 SUPPORT TASKS

GeneBench covers local-to-global genomic tasks comprehensively. For simplicity, we have segmented the GeneBench into short and long-range tasks based on a criterion of 1k length, considering that the sequence length significantly affects performance and complexity. The GeneBench benchmark encompasses diverse genomic targets, such as enhancers, promoters, and splice sites, at different scales. The tasks involve binary sequence classification, multi-class sequence classification, multi-class token classification, regression tasks, and mutation prediction on Coding DNA. A summary is shown in Table 2.

**Short-Range Tasks.** Short-range tasks are characterized by input lengths of less than one thousand. Our analysis covers thirty-eight datasets related to short-range tasks, which include various types of tasks like sequence classification, variant classification, Epigenetic mark prediction, promoter detection, enhancer prediction, transcription factor detection, and splice site prediction (Nguyen et al., 2024; Zhou et al., 2023; de Almeida et al., 2022; Ji et al., 2021).

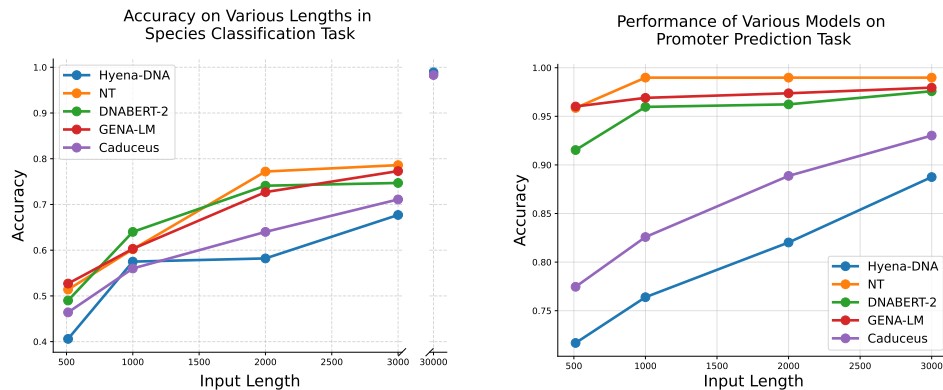

Figure 2: Impact of sequence length and models on accuracy. Left: Accuracy variation across different lengths in species. Right: Accuracy variation across different lengths in promoter prediction.

**Long-Range Tasks.** Long-range tasks are tasks with input lengths longer than 10K. Achieving state-of-the-art performance on benchmarks involving long sequences, such as the Long Range Arena (LRA) (Tay et al., 2020), is feasible. However, longer context lengths also introduce higher cost. For instance, attention-based models exhibit quadratic complexity concerning input length (Vaswani et al., 2017). In long-range tasks, we include site annotation (Jaganathan et al., 2019), species classification (Nguyen et al., 2024), promoter prediction (Fishman et al., 2023), chromatin profiling (Zhou & Troyanskaya, 2015), and genomic structure prediction (Schwessinger et al., 2020).

### 3.3 GENHYBRID MODEL

**Empirical Findings** As shown in Figure 2, through comprehensive benchmarking of current genetic foundation models, two key observations emerge that critically inform model design choices. **First**, models leveraging attention mechanisms, such as NT, DNABERT-2, and GENA-LM, consistently demonstrate superior performance on short-range sequence modeling tasks, particularly when input lengths are within the range of 500 to 3,000 tokens. This is evident across both species classification and promoter prediction tasks, where attention-based architectures excel at capturing intricate local dependencies. However, as sequence lengths increase beyond this range, the computational complexity of these models becomes a bottleneck, limiting their scalability. **Second**, models with linear time complexity, such as HyenaDNA and Caduceus, although initially underperforming in short-range tasks, exhibit robust accuracy when handling long sequences, with HyenaDNA achieving notable performance at lengths up to 30,000 tokens. Furthermore, attention-based models struggle to converge, while long-sequence models perform much superior in macro-level tasks, such as genome structure prediction. These findings underscore the trade-offs between modeling short-range versus long-range dependencies and reveal a need for hybrid architectures that can harness the strengths of both attention-based and SSMs. The detailed analysis refers to Sec. A.2.

**Mixing SSMs and Attentions Mechanisms** To address the identified trade-offs between short-range and long-range sequence modeling, we propose a simple yet effective hybrid architecture named Genhybrid that strategically incorporates two attention layers within an SSM-based model. Empirically, we find that replacing just two attention layers at the second layer and mid-level in the Caduceus leads to significant performance improvements, named GenHybrid-2. In addition, further introduction of full attention (GenHybrid-4) to long sequences instead causes negative effects (yellow line vs. gray line), as shown in Figure 3. It is worth noticing that the Transformer will OOM

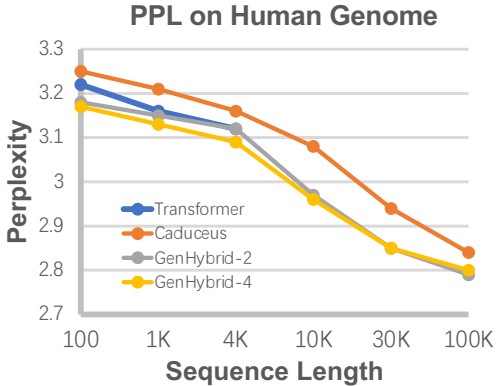

Figure 3: Pretraining on the hg-38 dataset.

at 30K sequence length. GenHybrid balanced approach allows the model to capture short-range dependencies efficiently at a critical stage without overwhelming the model's linear complexity benefits. As sequences grow longer, the remaining SSM-based layers ensure scalable and efficient processing. By introducing attention selectively, the hybrid model optimizes both computational efficiency and accuracy, excelling in a wide range of tasks, from short-range classification to long-range sequence prediction, and the detailed results are shown in Sec 4. This design offers a simple solution to the challenge of long sequence modeling, leveraging the best attention and SSM-based approaches with minimal overhead. The detailed description of GenHybrid is shown in Appendix B.2.

## 3.4 EVALUATION METRICS

We thoroughly assess the performance of the models supported for the tasks mentioned above by employing a range of metrics. These metrics are tailored to the specific characteristics of each task.

- **Error metrics:** We use cross-entropy to assess the variance between the anticipated outcomes and the actual targets in both binary and multi-class classification scenarios. On the other hand, Mean Squared Error (MSE) is applied in regression tasks.

- **Accuracy metrics:** We use top-1 accuracy for classification tasks and combine the evaluation metrics of computing the Area Under the Receiver Operating Characteristic Curve (AUC-ROC).

- **Correlation metrics:** Spearman correlation coefficient (Spearman) (Sedgwick, 2014) and the Pearson correlation coefficient (Pearson) (Kowalski, 1972) for regression tasks.

- **Computational metrics:** We utilize the number of parameters and the number of floating-point operations (FLOPs) to evaluate the computational complexity of the models.

## 3.5 CODEBASE STRUCTURE

Current open-source genomic foundation model codebases are typically constrained by a limited number of datasets and models. In contrast, GeneBench offers a versatile and expandable framework that follows the design principles outlined in HyenaDNA (Nguyen et al., 2024). We extend GeneBench to have a user-friendly interface, well-organized structure, and comprehensive content, thereby surpassing the usability of other open-source genomic foundational model codebases. For details, please refer to the Appendix A.

Table 3: Short-range tasks Top-1 accuracy (%) for pre-trained HyenaDNA, DNABERT, DNABERT2, GENA-LM, Nucleotide Transformer, Caduceus, and GenHybrid.

| Dataset | HyenaDNA(↑) | DNABERT(↑) | DNABERT2(↑) | GENA-LM(↑) | NT(↑) | Caduceus(↑) | GenHybrid(↑) | Initio Model(↑) |
|---|---|---|---|---|---|---|---|---|
| Mouse Enhancers | 0.7934 | 0.8099 | 0.8182 | 0.8297 | 0.8512 | 0.8163 | **0.8558** | 0.7008 |
| Coding vs Intergenomic | 0.9097 | 0.9364 | 0.9358 | 0.9324 | **0.9576** | 0.9372 | 0.9434 | 0.8844 |
| Human vs Worm | 0.9624 | 0.9584 | 0.9739 | 0.9698 | 0.9751 | 0.9557 | **0.9771** | 0.9408 |
| Human Enhancers Cohn | 0.7296 | 0.7023 | 0.7587 | 0.7563 | 0.7612 | 0.7376 | **0.7668** | 0.7080 |
| Human Enhancers Ensembl | 0.9033 | 0.8919 | 0.9075 | 0.9107 | 0.9244 | 0.8448 | **0.9281** | 0.7637 |
| Human Ensembl Regulatory | 0.8462 | 0.9380 | 0.8832 | 0.8810 | 0.9403 | 0.7367 | **0.9501** | 0.8616 |
| Human Nontata Promoters | 0.9445 | 0.8713 | 0.9524 | **0.9660** | 0.9295 | 0.8885 | 0.9567 | 0.8564 |
| Human OCR Ensembl | 0.7914 | 0.7496 | 0.7582 | 0.7898 | 0.8042 | 0.8176 | **0.8294** | 0.6947 |
| Human Core Promoter Detection | 0.8440 | 0.8491 | 0.8257 | 0.8140 | 0.8541 | 0.8505 | **0.8568** | 0.8003 |
| Human Transcription Factor Prediction | 0.6976 | 0.7840 | 0.8218 | 0.8240 | 0.8262 | 0.6928 | **0.8297** | 0.6672 |
| Human Promoter Detection | 0.7295 | 0.8393 | 0.8993 | 0.9001 | **0.9390** | 0.7322 | 0.9055 | 0.6875 |
| Human Splice Site Detection | 0.5660 | 0.8721 | 0.8813 | 0.9178 | **0.9481** | 0.5674 | 0.9272 | 0.5666 |
| Mouse Transcription Factor Prediction | 0.6535 | 0.7393 | 0.8269 | 0.8265 | 0.8502 | 0.6519 | **0.8513** | 0.6081 |
| Yeast Epigenetic Marks Prediction | 0.6301 | 0.7203 | **0.8022** | 0.7829 | 0.7845 | 0.6378 | 0.7998 | 0.6071 |
| Virus Covid Variant Classification | 0.3770 | 0.5990 | 0.7195 | **0.7033** | 0.6939 | 0.3794 | 0.5981 | 0.1974 |
| Central Dogma | 0.6720 | 0.6675 | 0.6834 | 0.6643 | 0.6803 | 0.6667 | **0.6880** | 0.6548 |

Table 4: Top-1 Pearson score for pre-trained HyenaDNA, DNABERT2, GENA-LM, Nucleotide Transformer, Caduceus, and GenHybrid and non-pre-trained model of deepstar in short task of drosophila enhancers prediction regarding the developmental (dev) and housekeeping activity (hk).

| Dataset | HyenaDNA(↑) | DNABERT2(↑) | GENA-LM(↑) | NT(↑) | Caduceus(↑) | DeepSTARR(↑) | GenHybrid(↑) |
|---|---|---|---|---|---|---|---|
| dev | 0.470 | 0.617 | **0.624** | 0.612 | 0.443 | 0.424 | 0.618 |
| hk | 0.552 | 0.734 | 0.740 | 0.736 | 0.530 | 0.513 | **0.742** |
| Mean | 0.511 | 0.678 | 0.682 | 0.674 | 0.486 | 0.468 | **0.688** |

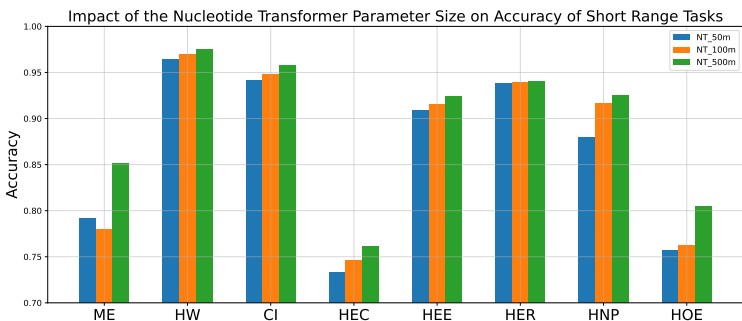

Figure 4: Evaluation of NT accuracy on short-range tasks with parameter sizes of 50M, 100M, and 500M. Task abbreviations: ME-Mouse Enhancers, HW-Human vs Worm, CI-Coding vs Intergenomic, HEC-Human Enhancers Cohn, HEE-Human Enhancers Ensembl, HR-Human Regulatory, HNP-Human Nontata Promoters, and HOE-Human OCR Ensembl.

## 4 EXPERIMENT AND ANALYSIS

We performed thorough experiments on the mentioned tasks to evaluate the effectiveness of the supported methods in GeneBench. The **bold** value indicates the best performance, and the underline value indicates the second-best performance. The Nucleotide Transformer is sometimes written NT. A detailed analysis of the results is provided to understand the genome foundation model better. For implementation specifics, please refer to the Appendix B.

### 4.1 SHORT RANGE TASKS

An experimental study was carried out to evaluate how different GFMs perform in handling short-range tasks. We draw several conclusions from the results. The details in Table 3 and Table 4.

**Except for the GenHybrid, pure convolution-based models consistently show lower performance compared to attention-based models, particularly on challenging tasks.** In binary classification scenarios, models like HyenaDNA show a notable decrease in accuracy, with reductions of approximately 0.053 compared to attention-based counterparts. This discrepancy becomes even more pronounced in multi-class classification tasks, where the accuracy gap widens to around 0.239. Similarly, Caduceus demonstrates a comparable pattern, with an accuracy gap of about 0.053 in binary classification tasks and a significantly larger margin of 0.274 in multi-class classification assignments. When comparing these models to pre-trained models, the discrepancy becomes even more striking. The CNN model, our initio model without any pretraining, in particular, achieves substantially lower accuracy. For the tasks of drosophila enhancers prediction, HyenaDNA achieves a Pearson score of 0.470, notably lower than attention-based models like GENA-LM and DNABERT2, which score 0.624 and 0.617, respectively. Caduceus records an even lower score of 0.443 in this category. In the housekeeping activity dataset, HyenaDNA and Caduceus obtain scores of 0.552 and 0.530, while attention-based models like GENA-LM and DNABERT2 achieve higher scores of 0.740 and 0.734. On average, GenHybrid outperforms all models with a mean Pearson score of 0.688, surpassing even the top-performing attention-based models. The non-pre-trained model DeepSTARR records the lowest mean score of 0.468. Most interestingly, this class of mutation prediction is not done well for current GFMs, and this is really a direction to focus on in the future. The detailed results are shown in the Appendix C.

**The size of parameters plays a crucial role in determining performance**. Similar to NLP, the scaling law works in short-range tasks in Figure 4. Among the models analyzed, the Nucleotide Transformer, which boasts the largest parameter size, outperformed others on 11 out of 15 datasets. Subsequently, GENA-LM and DNABERT2, having similar parameter sizes, excelled on the 2 and 1 datasets, respectively. Noteworthy is the performance disparity exhibited by DNABERT compared to other attention-based models, with variances ranging from 0.0050 to 0.1401, despite sharing a similar architecture with the Nucleotide Transformer, albeit possessing the smallest parameter size. To delve deeper into the impact of parameter size, we examined the performance of Nucleotide Transformers with parameter sizes of 50, 100, and 500 million.

## 4.2 LONG RANGE TASKS

**Pure Attentions collapse on Genome Structure Prediction.** The results of long-range tasks are detailed in Table 5, 6, 7, and 8 and Appendix C shows the corresponding visualizations. In a nutshell, the simple GenHybrid still shows its superiority. In Genomic Structure prediction, surprisingly, attention-based models fail to converge in this "longest task" and utilizing a pre-trained model as a backbone does not show significant benefits, as Orca achieved the second-best Pearson and MSE in H1-ESC and the best Pearson and MSE in HFF, respectively. We can find a greater resemblance to image data when nucleotide sequences become longer: short vocabulary (4 bases ATCG and 256 RGB values) and fixed data rules (chromosome and image content).

**As the sequence grows, the SSM/convolution-based models become more efficient.** In other long-range tasks, the difference between attention-based and convolution-based models has been significantly reduced. For Species Classification and Promoters Prediction, DNABERT2, GENA-LM, and Nucleotide Transformer exhibit comparable performance, while HyenaDNA performs notably worse than the rest. However, in the task of splice site annotation, Caduceus achieves the second highest performance, with no significant performance gap between them. The computational overhead of the convolutional model is much smaller than that of the attention-based model for the same sequence length, as shown in Figure 6. Extra vertical model comparison on different scales is shown in Appendix C. We can observe that when the model size is smaller, and the model entirely on Attention is worse than the same configuration on 32k sequence length.

## 4.3 GENE CLUSTERING

In this section, we examine the fine-tuned embedding models HyenaDNA, DNABERT2, GENA-LM, Caduceus, Nucleotide Transformer, and GenHybrid. These models are utilized to encode gene

Table 5: Top-1 Pearson and MSE for pre-trained Orca, HyenaDNA, DNABERT2, Caduceus, Nucleotide Transformer, and GenHybrid in the long-range task of Genomic Structure Prediction.

| Dataset | Orca | | HyenaDNA | | Caduceus | | DNABERT2 | | NT | | GenHybrid | |
|---|---|---|---|---|---|---|---|---|---|---|---|---|
| | Pearson(↑) | MSE(↓) | Pearson(↑) | MSE(↓) | Pearson(↑) | MSE(↓) | Pearson(↑) | MSE(↓) | Pearson(↑) | MSE(↓) | Pearson(↑) | MSE(↓) |
| H1-ESC | 0.4543 | 0.0175 | 0.4357 | 0.0184 | 0.5024 | 0.0168 | - | - | - | - | **0.5189** | **0.0158** |
| HFF | 0.4350 | 0.0911 | 0.3103 | 0.1013 | 0.3536 | 0.1047 | - | - | - | - | **0.4531** | **0.0897** |

Table 6: Top-1 Spearman and MSE for pre-trained HyenaDNA, DNABERT2, Caduceus, and GenHybrid in long-range task of Bulk RNA Expression.

| Dataset | HyenaDNA | | Caduceus | | DNABERT2 | | NT | | GenHybrid | |
|---|---|---|---|---|---|---|---|---|---|---|
| | Spearman(↑) | MSE(↓) | Spearman(↑) | MSE(↓) | Spearman(↑) | MSE(↓) | Spearman(↑) | MSE(↓) | Spearman(↑) | MSE(↓) |
| Bulk | 0.737 | 0.517 | 0.738 | 0.512 | 0.748 | 0.483 | 0.755 | 0.463 | **0.768** | **0.452** |

**Sequence Embeddings, Colored by Species**

Figure 5: The t-SNE visualization of DNA embedding for foundation model in species classification. Including embedding for DNABERT2 with an accuracy of 0.742, embedding for HyenaDNA with an accuracy of 0.655, embedding for the NT with an accuracy of 0.761, embedding for Caduceus with an accuracy of 0.703, and embedding for GenHybrid with an accuracy of 0.772.

Table 7: Top-1 AUC-ROC Score for pre-trained HyenaDNA, DNABERT2, GENA-LM, Nucleotide Transformer, Caduceus, and GenHybrid in the long-range task of splice site prediction.

| Dataset | HyenaDNA(↑) | DNABERT2(↑) | GENA-LM(↑) | NT(↑) | Caduceus(↑) | SpliceAI(↑) | GenHybrid(↑) |
|---|---|---|---|---|---|---|---|
| Splice donar | 0.574 | 0.635 | 0.629 | 0.557 | 0.642 | 0.574 | **0.653** |
| Splice acceptor | 0.723 | 0.707 | 0.730 | 0.722 | 0.740 | 0.691 | **0.752** |
| Mean | 0.648 | 0.671 | 0.679 | 0.639 | 0.691 | 0.632 | **0.700** |

Table 8: Top-1 accuracy for pre-trained HyenaDNA, DNABERT2, GENA-LM, NT, Caduceus, and GenHybrid in long-range tasks of Species Classification and Promoters Prediction.

| Dataset | HyenaDNA(↑) | DNABERT2(↑) | GENA-LM(↑) | NT(↑) | Caduceus(↑) | GenHybrid(↑) |
|---|---|---|---|---|---|---|
| Species Classification | 0.6550 | 0.7420 | 0.7430 | 0.7610 | 0.7030 | **0.7720** |
| Promoters Prediction | 0.8875 | 0.9758 | 0.9795 | **0.9890** | 0.9302 | **0.9890** |

sequences from various species. To visualize the embeddings, we extract the representations from the final hidden layer of each model and apply t-distributed Stochastic Neighbor Embedding (t-SNE) propose in Van der Maaten & Hinton (2008). The visualization, presented in Figure 5, reveals clear clusters that offer both visual and quantitative insights. For instance, the propose GenHybrid, which demonstrates the highest accuracy among the models, shows well-separated embeddings for distinct species, indicating effective differentiation. In contrast, HyenaDNA, which has the lowest accuracy, displays less differentiation among the embeddings of different species, suggesting that its representations are less distinct. This visualization underscores the varying capabilities of distinguishing between gene sequences from different species, with NT and GenHybrid excelling in accuracy and clarity of separation, while HyenaDNA struggles in comparison. From the results, it is clear that the k-mer based approaches have a more significant advantage.

## 4.4 COMPUTATIONAL COST

Being able to handle long sequences is a critical step in GFM. Therefore, we compared the floating-point operations per second (FLOPS) as a metric to evaluate the computational efficiency of each model relative to the various input lengths, as shown in Figure 6. Typically, attention-based models demonstrate significantly higher computational capabilities, followed by attention-free foundational models. Simple CNN models, on the other hand, exhibit the lowest computational cost. Regarding computational efficiency, GenHybrid demonstrates its outstanding efficiency among these models.

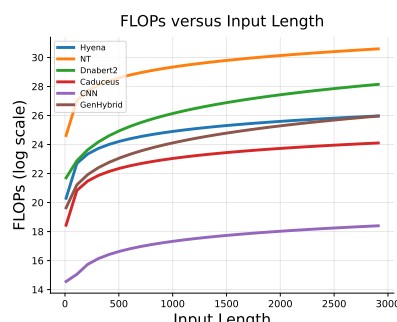

Figure 6: Flops versus input length

## 5 CONCLUSION AND DICUSSION

This paper presents GeneBench, a comprehensive benchmark for GFMs featuring ten representative models covering a broad spectrum of challenging tasks from local to global view of genomics. GeneBench classifies existing approaches into attention-based and convolution-based GFMs. Extensive experiments are carried out to systematically assess the performance of the models supported across various tasks. In short-range tasks, attention-based models excel at capturing intrinsic information, while attention-free models achieve comparable yet less efficient performance. In long-range tasks, the performance difference between attention-based and convolution-based models becomes narrower. Furthermore, increasing input length can significantly enhance performance on conv/SSM models. Based on our experimental results, we, therefore, propose GenHybrid, a simple yet efficient model co-designed by SSM and attention to performing better on all genetic tasks we covered. **Limitations.** Despite the multifaceted comparisons of GFMs, GeneBench is basically stuck on downstream task prediction, and comparisons on pre-training are lacking. For example, the impact of pre-training data under different model structures, *etc.* In addition, we have not verified the performance of GFM on the whole chromosome due to the limitation of computational resources.

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

## A  CODEBASE OVERVIEW

In this section, we present a comprehensive overview of the codebase structure of GeneBench. The codebase is organized into three abstracted layers, namely the core layer, algorithm layer, and user interface layer, arranged from the bottom to the top, as illustrated in Figure 7. Our codebase is under Apache-2.0 license, like HyenaDNA Nguyen et al. (2024).

**Core Layer**  The core layer of GeneBench includes key elements like data loaders for supported datasets, fundamental modules for supported models, and metrics for evaluation. Data loaders provide a standardized way of loading and preprocessing data. The modules contain essential unit implementations of supported models. Metrics offer a consistent method for evaluating results. This core layer sets the groundwork for upper layers to ensure adaptable usage.

**Algorithm Layer**  The algorithm layer encompasses the implementations of the supported models, which are divided into two main categories: attention-based and convolution-based models. These implementations are developed using the PyTorch framework and closely adhere to the methodologies described in the original research papers and their official open-source code. To enhance convenience, we directly incorporate the pretrained model from Huggingface. The algorithm layer ensures the compatibility, reliability, and reproducibility of the supported algorithms by abstracting common elements and preventing code duplication, thus facilitating easy and flexible integration of customized algorithms. Moreover, this layer provides a standardized interface that simplifies tasks such as model training, evaluation, and testing. By offering a consistent interface, the algorithm layer enhances user-friendliness and promotes seamless experimentation with the models.

**User Interface Layer**  The user interface layer includes configurations, training, Experiments, and scripts to support the basic functions of GeneBench. It provides user-friendly tools for creating

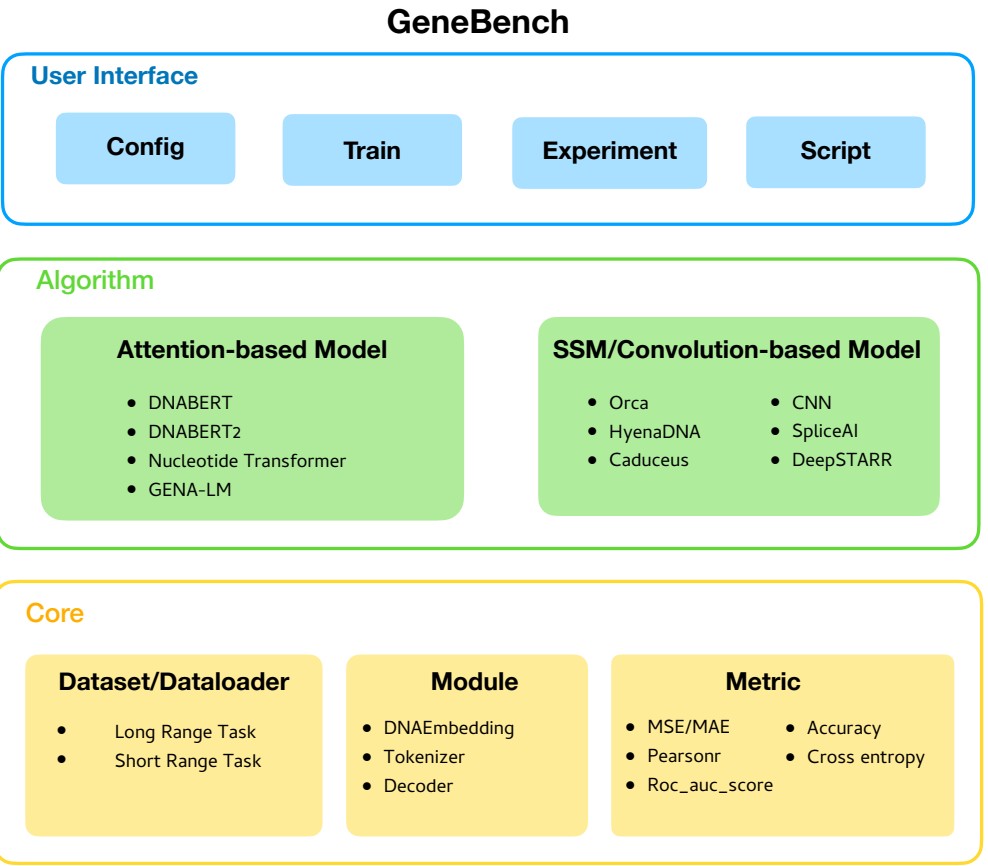

Figure 7: The graphical overview of GeneBench.

visualizations. This layer is designed to be intuitive, allowing users to easily train, evaluate, and test the algorithms it supports. Through detailed parameter settings in the configurations, the user interface layer offers a unified interface that enables users to replicate the results presented in this paper without the need for extra steps.

## A.1 DETAILED DATA DESCRIPTION

*Mouse Enhancers Ensembl*, *Coding vs Intergenomic*, *Human vs Worm*, *Human Enhancers Cohn*, *Human Enhancers Ensembl*, *Human Ensembl Regulatory*, *Human Nontata promoters*, and *Human OCR Ensembl* were referenced from Grešová et al. (2023). In this context, "Human" and "Mouse" signify the origin of the genetic sequences, while "Enhancers," "Regulatory," "OCR," and "promoter" describe the nature of the sequences. A regulatory gene is responsible for controlling the expression of one or more structural genes. Enhancers are specific genomic elements that regulate gene expression without requiring close proximity to the target gene. Open chromatin regions (OCR) are parts of the genome that can be easily accessed by DNA regulatory elements. On the other hand, a promoter is a segment within a gene where specific proteins bind to initiate the gene's transcription. The term "Ensembl" in this context refers to the data resources provided by The Ensembl project (Howe et al., 2021).

The tasks of *Human Promoter Prediction*, *Human Core Promoter Detection*, *Human Transcription Factor Prediction*, *Human Splice Site Detection*, *Mouse Transcription Factor Prediction*, *Yeast Epigenetic Marks Prediction*, and *Virus Covid Variant Classification* adopted from Zhou et al. (2023) encompass a variety of objectives. These tasks involve predicting different region types, such as promoters, transcription factor binding sites, and splice sites across multiple animal genes, as well as predicting variants of the Covid virus based on provided gene sequences.

We also include *Splice Site Prediction*, *Promoter Prediction*, and *Drosophila Enhancer Detection* in our assessment following the methodology described in Fishman et al. (2023). These datasets are known for their extensive sequences and varied tasks. They comprise sequences exceeding 1000 base pairs, covering a range of tasks like token classification, sequence-level classification, and regression. In particular, *Drosophila enhancers prediction* involves a two-class regression, where the goal is to predict two float values for every 249-base pair sequence, one for housekeeping and one for developmental enhancer scores.

More importantly, we introduce short-range task Central Dogma and long-range task *genomic structure prediction*. These predictions specifically examines how the transferability in multi-omics and structural variants impact genome organization at various scales. Additionally, we have included *Species Classification* from Nguyen et al. (2024), which has heightened the complexity of classification by encompassing a larger number of species. We incorporated the task of *Bulk RNA expression* to evaluate the model's performance within a lengthy context (Kao et al., 2024). Bulk RNA sequencing is a biological assay that gauges the average gene expression from a group of cells.

## A.2 EFFECT OF SEQUENCE LENGTH

Recall the results in Figure 2. We conducted an analysis to evaluate the impact of sequence length on model performance in long-range genomic tasks, that is also the motivation why we propose GenHybrid. Specifically, we used input sequences of varying lengths—512, 1000, 2000, and 3000 base pairs (bp)—to assess the performance of four models: Hyena-DNA, Nucleotide Transformer, DNABERT-2, and GENA-LM in both species and promoter prediction tasks. Additionally, to explore the potential of convolution-based models, we tested a significantly longer input sequence of 30,000 bp with Hyena-DNA, focusing exclusively on the species prediction task. From the data, it is evident that increasing the sequence length consistently enhances performance across all the models tested. This trend is particularly pronounced with Hyena-DNA, which, despite trailing behind attention-based models at shorter context lengths, exhibits superior performance with longer contexts. This improvement underscores the advantages of using extended context lengths in genomic sequence analysis. However, this benefit is not without its challenges. In tasks like promoter prediction, where input length is inherently capped at 3000 bp, Hyena-DNA's reliance on longer sequences becomes a limiting factor. This limitation presents a significant area for future research, aiming to optimize model performance within these constraints and potentially develop novel approaches to leverage longer sequences more effectively within the confines of specific genomic datasets.

## B  IMPLEMENTATION DETAILS

The table presented in Table 9 outlines the hyperparameters utilized in the various models supported across different datasets. Each model's hyperparameters consist of layers, a width of the hidden dimension, parameter size, learning rate, embed dropout, residual dropout, optimizer, optimizer momentum, training epochs, batch size, LR scheduler, and reverse complement augmentation. It is important to note that sequence length is task-dependent and not directly related to the model.

### B.1  MODEL DESCRIPTION

**DNABERT** Ji et al. (2021) represents the pioneering deep learning approach that incorporates the concept of Bidirectional Encoder Representations from Transformers (BERT) model (Devlin et al., 2018) within the context of genomic DNA. Similar to BERT, DNABERT follows a pre-training—fine-tuning framework. In the pretraining phase, a portion of k contiguous tokens, covering 15% of the sequence, is randomly masked, prompting DNABERT to forecast the masked sequences based on the remaining context. The training dataset is derived from the human genome using a direct non-overlapping splitting and random sampling approach, with sequence lengths ranging from 5 to 510.

**Nucleotide Transformer** utilizes an encoder-only transformer architecture. The models are trained using the BERT methodology. The Nucleotide Transformer employs three distinct datasets for pre-training the model: The Human reference genome dataset, The 1000G dataset, and The Multispecies dataset (Dalla-Torre et al., 2023).

**DNABERT-2** utilizes the Transformer Encoder architecture, providing flexibility in input length and enhanced computational and memory efficiency. It replaces learned positional embeddings with Attention with Linear Biases (ALiBi) (Press et al., 2021) and incorporates FlashAttention (Dao et al., 2022) and Low Precision Layer Normalization. The model is pretrained on The Human Genome dataset and The Multi-Species Genome dataset (Zhou et al., 2023).

**GENA-LM** model utilizes the Transformer Encoder architecture and has been trained on the Human T2T v2 genome assembly dataset.

**Hyena-DNA** utilizes a decoder-only design, composed of a series of blocks containing a Hyena operator. It is pretrained using the human reference genome.

**Caduceus** is a group of bidirectional long-range DNA sequence models that are the pioneers in supporting RC equivariant language modeling. Caduces employ pre-training and fine-tuning techniques with MambaDNA as their foundation.

**GenHybrid** is a hybrid model that strategically incorporates two attention layers within an SSM-based model. In our case, we employed Caduceus as the baseline and replaced the second layer and the middle layer with full attention. The training procedure is the same as Caduceus.

The convolution-based deep learning models such as **CNN**, **SpliceAI**, **DeepSTARR**, and **Orca** are specifically developed to predict distinct genomic features. These models are trained from scratch using specialized datasets instead of being pretrained on general genomic sequences.

### B.2  MODULE DESCRIPTION

**Attention** is the scaled dot product operation used to represent the relationships within the input or output sequence. This attention mechanism plays a crucial role in the Transformer model, which has been a significant advancement in deep learning (Devlin et al., 2018; Radford et al., 2018). The formulation of attention is as follows:

$$\text{Attention}(Q, K, V) = \text{softmax}\left(\frac{QK^T}{\sqrt{d_k}}\right) V \tag{3}$$

Where $Q$, $K$, and $V$ are mapped from the input with linear layer.

**Hyena** a class of data-controlled operators that involve a combination of multiplicative gating interactions and long convolutions, introduced by Poli et al. (2023). The formulation of attention is as follows:

$$y = \text{H}(u)v = \text{D}_x^N \, \text{S}_h^N \cdots \text{D}_x^2 \, \text{S}_h^2 \text{D}_x^1 \, \text{S}_h^1 v \tag{4}$$

Table 9: Hyperparameter ranges used to fine-tune all models for all datasets.

| Configuration | HyenaDNA | DNABERT | DNABERT2 | GENA-LM | NT | Caduceus | GenHybrid |
|---|---|---|---|---|---|---|---|
| Layers | 8 | 12 | 12 | 12 | 29 | 16 | 10+2 |
| Width | 256 | 768 | 768 | 768 | 768 | 1024 | 256 |
| Parameters | 6.6 M | 86.1M | 117M | 113M | 498M | 7.9M | 79.5M |
| Optimizer | | | | AdamW | | | |
| Optimizer momentum | | | | $\beta_1, \beta_2 = 0.9, 0.999$ | | | |
| Training epoch | | | | 100 | | | |
| Batch size | | | | 128-256 | | | |
| Learning rate | 1e-4 to 6e-4 | 3e-5 | 3e-5 | 5e-5 | 1e-5 | 1e-4 to 1e-3 | 3e-5 to 1e-5 |
| LR scheduler | | | | Cosine decay | | | |
| Weight decay (model) | 0.1 | 0.1 | 0.1 | 0.1 | 0.1 | 0.1 | 0.1 |
| Weight decay (Hyena layers) | 0 | \ | \ | \ | \ | \ | \ |
| Embed dropout | 0.1 | 0.1 | 0.1 | 0.1 | 0.0 | 0.0 | 0.0 |
| Resid dropout | 0.0 | 0.0 | 0.0 | 0.0 | 0.0 | 0.0 | 0.0 |
| Reverse complement aug. | | | | False | | | |
| Sequence lengths | 30 to 30k | 30 to 512 | 30 to 3k | 30 to 3k | 30 to 3k | 30 to 30k | 30 to 30k |

Where $\mathrm{D}_x^n = \mathrm{diag}\,(x^n) \in \mathbb{R}^{L \times L}$ and $\mathrm{S}_x^n$ are Toeplitz matrix corresponding to $\mathrm{h}^n$ (Farenick, 2021).

**State Space Model** is an class of sequence models have proven to be effective at handling long-range models (Gu & Dao, 2023). The formulation of attention is as follows:

$$\dot{\boldsymbol{h}}(t) = \boldsymbol{A}h(t) + \boldsymbol{B}x(t), \quad y(t) = \boldsymbol{C}h(t) + \boldsymbol{D}x(t) \tag{5}$$

Where $\boldsymbol{A} \in \mathbb{R}^{N \times N}$, $\boldsymbol{B} \in \mathbb{R}^{N \times 1}$, $\boldsymbol{C} \in \mathbb{R}^{1 \times N}$, and $\boldsymbol{D} \in \mathbb{R}$ are the parameters of the system.

# C ADDITIONAL RESULT

**Genomic Structure Prediction** In addition to the quantitative results presented in the main text, we also offer a visual representation for qualitative evaluation, as depicted in Figure 8 and Figure 9. Across all models examined, we illustrate both the accurate and inaccurate prediction outcomes for comparison. It is noted that while HyenaDNA and DNABERT2 exhibit diverse predictions, Orca's predictions are relatively consistent. GenHybrid predicts the most relevant results with the ground truth. Moreover, we also visualized the **Drosophila Enhancer Detection** task in Figure 10. The visualization results of **Bulk RNA Expression** tasks are in Figure 11.

**Model Vertical Comparison** we set different hidden sizes of the models to draw the compute-performance table below separately. Specifically, we do pre-training based on MLM on the human genome hg-38 with sequence lengths of 2k and 32k, respectively. We take the eval NTK loss of the 10000th step to do the vertical evaluation of the model architecture. The range of hidden state size in 128, 256, 512 and the number of layers is fixed as 6. We can observe that when the hidden size is smaller and the model based entirely on Attention is instead not as good as the 2k model with the same configuration on 32k sequence length, which also verifies that the findings of our paper are mutually verifiable.

Table 10: The PPL results of MLM pre-training on hg-38 with sequence lengths of 2k.

| | NT | DNABERT-2 | HyenaDNA | Caduceus | GenHybrid |
|---|---|---|---|---|---|
| 128 | 1.141 | 1.138 | 1.147 | 1.118 | 1.114 |
| 256 | 1.102 | 1.088 | 1.121 | 1.067 | 1.062 |
| 512 | 1.058 | 0.989 | 1.011 | 0.993 | 0.990 |

Table 11: The PPL results of MLM pre-training on hg-38 with sequence lengths of 32k.

| | NT | DNABERT-2 | HyenaDNA | Caduceus | GenHybrid |
|---|---|---|---|---|---|
| 128 | 1.221 | 1.144 | 1.112 | 1.101 | 1.092 |
| 256 | 1.102 | 1.078 | 1.085 | 1.042 | 1.024 |
| 512 | 0.992 | 0.984 | 0.989 | 0.982 | 0.980 |

**Mutation Prediction on Coding DNA** Zero-shot mutation prediction on coding DNA sequence (protein): We collected deep mutational scanning (DMS) datasets (Livesey & Marsh, 2023), where numerous mutations are introduced into a coding region, and each variant is assigned an experimentally determined fitness score. The specific data flow is as follows: First, define a list of mutation information, where each entry contains the mutation type and its related attributes. Next, convert the data into input strings that include a mask for easier processing by the model. Then, load the pre-trained genomic model and set it to evaluation mode. Following this, extract the scores at the masked positions using the logits output by the model, and return the highest-scoring predicted words to identify the mutation sites. Additionally, calculate the overall likelihood of the input text by applying softmax to the logits and summing the log likelihoods. Finally, for each mutation text, invoke the prediction and likelihood calculation functions to output the corresponding prediction results and likelihood scores. The results are listed in Table 13.

## D  FORTY-FOUR DATASETS

**Comprehensive Datasets** We count sub-datasets as individual datasets; for instance, Human Transcription Factor Prediction includes five sub-datasets (see Table 10). The total size is Human Core Promoter Detection (3) + Human Transcription Factor Prediction (5) + Human Promoter Detection (3) + Human Splice Site Detection (1) + Mouse Transcription Factor Prediction(5)+ Yeast Epigenetic Marks Prediction (10) + Virus Covid Variant Classification (1) + Mouse Enhancers (1) + Coding vs Intergenomic (1) + Human vs Worm (1) + Human Enhancers Cohn (1) + Human Enhancers Ensembl (1) + Human Ensembl Regulatory (1) + Human Nontata promoters (1) + Human OCR Ensembl (1) + Drosophila Enhancers Prediction (1) +Splice Site Prediction (1) + Species Classification (1) + Promoters Prediction (1) + Genomic Structure Prediction(2) + Bulk RNA Prediction (1) + Central Dogma (1) + Mutation Prediction on Coding DNA (1)

**Details of Proposed Datasets**

- Genome Structure Prediction: We utilized micro-C datasets from H1-ESCs and HFF cells, obtained from the 4D Nucleome portal, along with genomic sequences from GRCh38/hg38. Using the Selene library, we implemented on-the-fly training data generation, sampling uniformly from training chromosomes. For each micro-C dataset, we trained a separate model. Training samples consisted of input sequences paired with their corresponding genome interaction matrices, which were derived from normalized contact matrices. To prepare the genome interaction matrices, we applied two key procedures to the micro-C datasets using cooler and cool tools packages: iterative correction matrix balancing and adaptive coarse-graining. The latter smooths sparse areas of the contact map using variable window sizes to eliminate zeros. We maintained spatial resolution by avoiding additional smoothing. The final processed matrix was normalized using a background matrix derived from distance encoding, with adjustments for numerical stability. We computed distance-based expectations per chromosome using cool tools, aggregated them across chromosomes, and applied lowess smoothing for distances beyond 1.6 Mb. For model evaluation, we split the chromosomes into three sets: training (all except chr8, 9, and 10), validation (chr8), and testing (chr9, 10).

- Coding DNA Mutation Prediction: We predict gene mutation loci through a masked learning approach. The specific data flow is as follows: First, define a list of mutation information, where each entry contains the mutation type and its related attributes. Next, convert the data into input strings that include a mask for easier processing by the model. Then, load the pre-trained genomic model and set it to evaluation mode. Following this, extract the scores at the masked positions using the logits output by the model, and return the highest-scoring predicted words to identify the mutation sites. Additionally, calculate the overall likelihood of the input text by applying softmax to the logits and summing the log-likelihoods. Finally, for each mutation text, invoke the prediction and likelihood calculation functions to output the corresponding prediction results and likelihood scores.

**Comparison with Other benchmarks** As shown in Table 12, GeneBench stands out compared to other genomic benchmarks, such as BEND, Genomic Benchmarks, and GUE, by providing a much

Table 12: Benchmarks comparison

|  | GeneBench(ours) | BEND (Marin et al., 2023) | Genomic Benchmarks (Grešová et al., 2023) | GUE (Zhou et al., 2023) |
|---|---|---|---|---|
| Number of Datasets | 45 | 6 | 9 | 28 |
| Long range tasks | yes | yes | no | no |
| Regression tasks | yes | no | no | no |
| New method proposed | yes | no | no | no |

Table 13: The results of Spearman correlation on mutation prediction on Coding DNA sequence.

|  | NT | DNABERT-2 | HyenaDNA | Caduceus | GenHybrid |
|---|---|---|---|---|---|
| Spearman correlation | 0.14 | 0.12 | 0.15 | 0.18 | 0.18 |

more comprehensive evaluation framework. It includes 45 datasets, which is significantly more than the others, and supports both long-range and regression tasks, making it suitable for a wider variety of genomic modeling challenges. Additionally, GeneBench is the only benchmark that proposes a new method, demonstrating its contribution not only in benchmarking but also in advancing the field of genomic modeling. These features make GeneBench a more versatile and robust choice for evaluating genomic foundation models.

**Discussion** We found that current DNA language models primarily fall into two categories: CNN-based and attention-based pre-trained models. While the incorporation of attention mechanisms can help us better understand deoxyribonucleotide sequences, convolutional models outperform attention models in certain specific tasks. Therefore, it is particularly important to combine attention and convolutional modules, as this not only aids in a more comprehensive understanding of both long and short genomic tasks but also effectively alleviates the high computational costs associated with the pre-training phase. To this end, we propose GenHybrid, a simple yet efficient model co-designed with SSM and attention, which achieves state-of-the-art performance across multiple tasks.

In short-range tasks, attention-based models excel at capturing intrinsic information, while attention-free models perform comparably but less efficiently. In long-range tasks, the performance difference between attention-based and convolutional models becomes narrower. Moreover, increasing input length can significantly enhance performance, especially in extending gene context. Based on our experimental results, we have developed GenHybrid, a model that integrates SSM and attention to perform better on all genetic tasks we covered.

It is noteworthy that we introduced central rules and DNA structural prediction data to evaluate the performance of existing models. The inclusion of these two evaluation datasets allows us to assess genomic models not only on the basis of single genomic sequence information but also incorporating DNA structural information and DNA-protein interactions. Furthermore, we found that existing models are not effective in handling genomic mutation data, indicating that future research should focus on further optimizing model design for this task.

**Limitations and Outlook** Considering that current genomic pre-training models are primarily developed using human data, we mainly evaluated the performance of genomic pre-training models on 45 human downstream tasks in GenBench. However, the genetic sequence differences among biologically distinct species are significant, and designing a universal pre-training model would better serve the diverse downstream tasks. Additionally, we are unclear about how these models learn biological sequence information during the pre-training phase; assessing the interpretability of the models will help us better design and understand them, thereby supporting "precision medicine." To further model the structural information of genomes, we should also consider fine-grained details such as DNA bond angles, bond energies, and bond lengths, as well as the physicochemical properties of deoxyribonucleotides and their impact on the models.

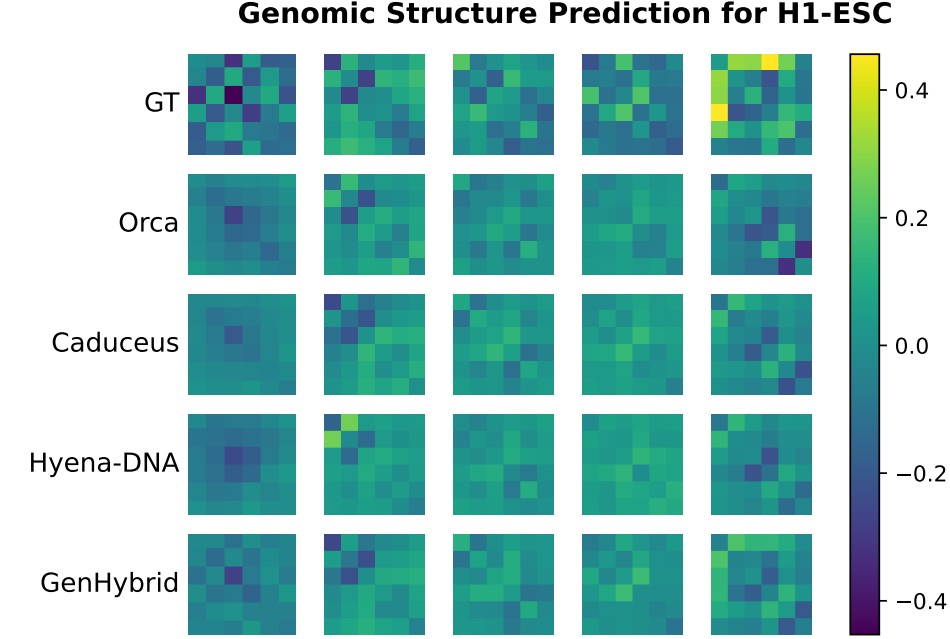

Figure 8: The results visualization of Genomic structure predictions from Orca, HyenaDNA, Caduceus, and GenHybrid (y-axis) in H1-ESC with the batch size is five (x-axis).

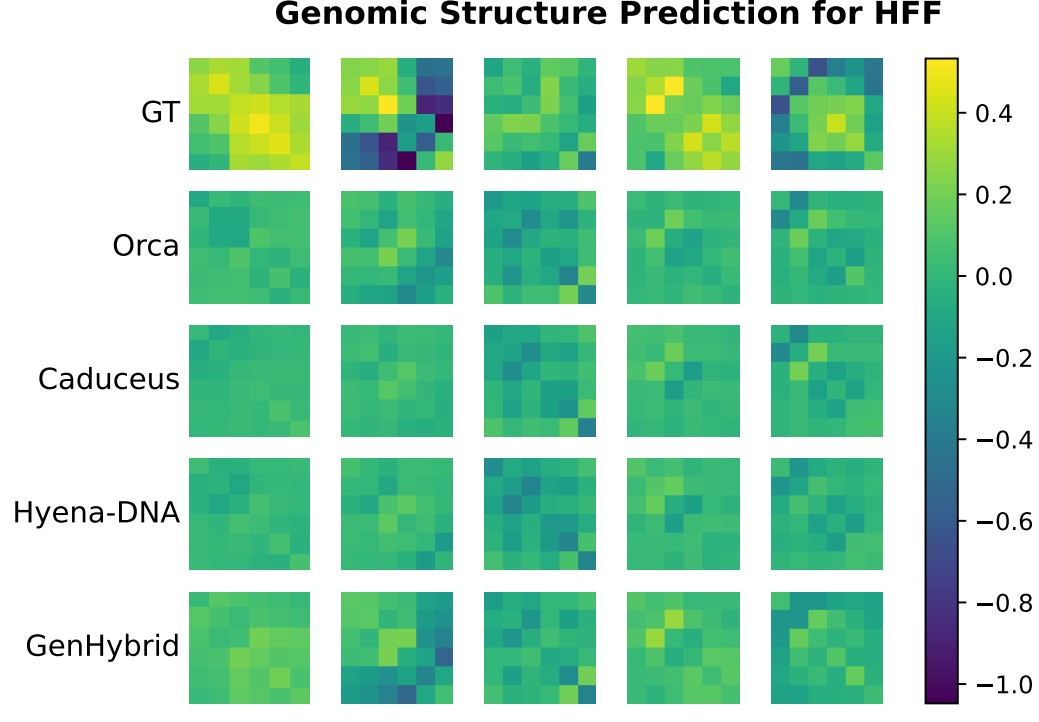

Figure 9: The results visualization of Genomic structure predictions from Orca, HyenaDNA, Caduceus, and GenHybrid (y-axis) in Hff with the batch size as 5 (x-axis).

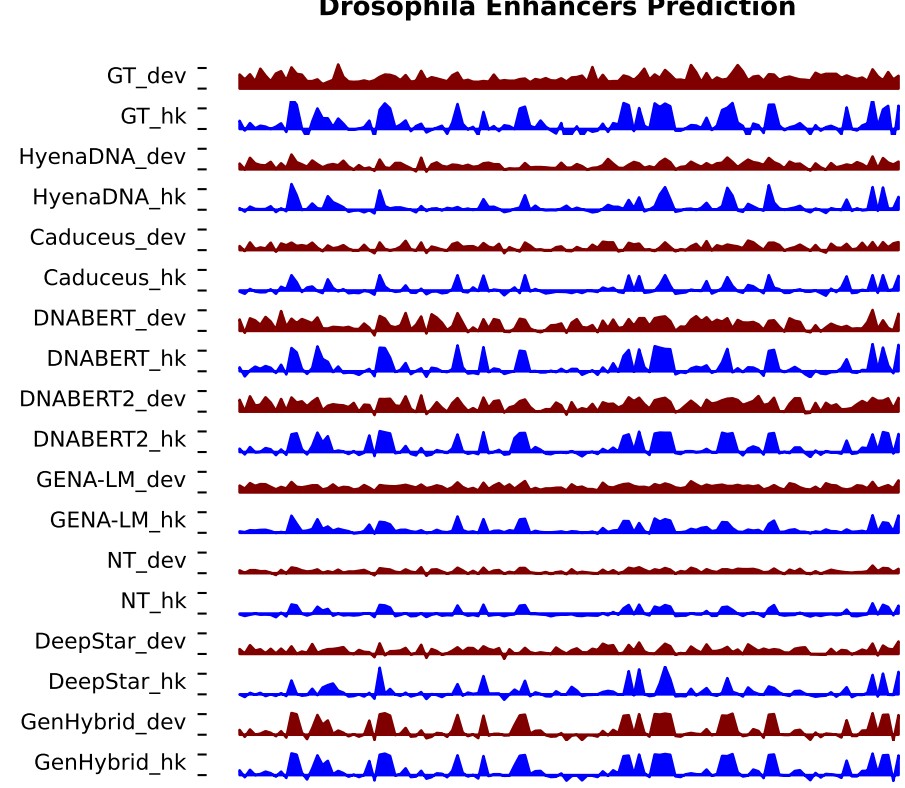

Figure 10: We present the visualization of Drosophila Enhancer Detection for HyenaDNA, Caduceus, DNABERT, GENA-LM, NT, and DNABERT2. The visualization illustrates the scores for both housekeeping and developmental enhancers with a batch size of 128 (x-axis). Additionally, we include the calculation of the Pearson correlation coefficient with the actual data on the right side.

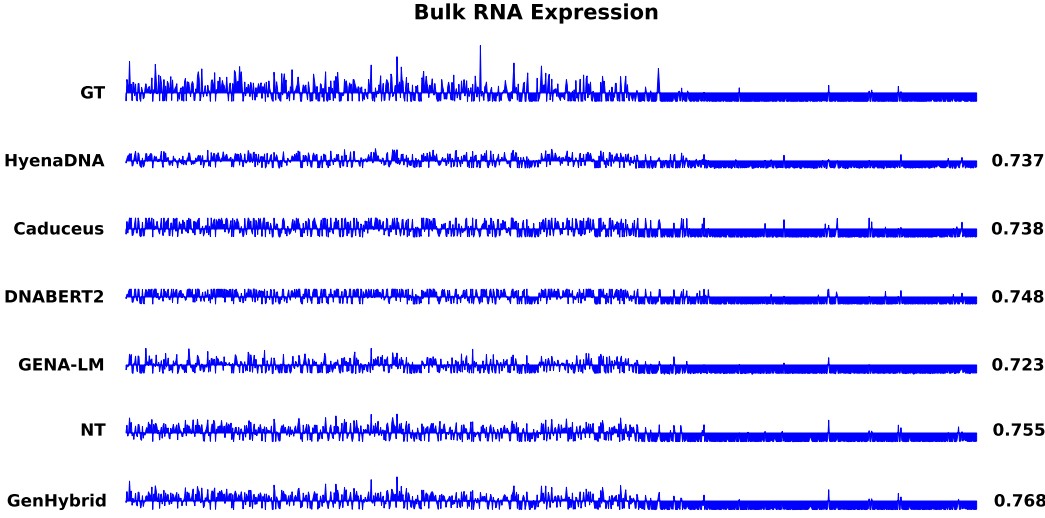

Figure 11: We present the visualization of Bulk RNA Expression for HyenaDNA, Caduceus, DNABERT, GENA-LM, NT, and DNABERT2. The visualization illustrates the expression levels in tissue type 0. Additionally, we include the calculation of the Spearman correlation coefficient with the actual data on the right side. The max length is fixed as 2048 (x-axis).

