# OpenReview forum: "GeneBench: Systematic Evaluation of Genomic Foundation Models and Beyond"
_ICLR.cc/2025/Conference — ICLR 2025 Conference Withdrawn Submission_

### Official Review · Reviewer_u9EG · 2024-11-01

**Soundness:** 3
**Presentation:** 3
**Contribution:** 3
**Rating:** 5
**Confidence:** 3

**Summary:**

This study introduces a comprehensive benchmark suite, GeneBench, for evaluating the efficacy of Genomics Foundation Models. The authors systematically evaluated several DNA tasks including coding region, non-coding region, and genome structure. They also provided some insights into the model design and model training.

**Strengths:**

The manuscript is well-organised and the experiments are relatively comprehensive.

**Weaknesses:**

Lack of biological insights

**Questions:**

1. The authors are suggested to state the advantages of their method. For example, why a bioinformatian should use their proposed method instead of others?
2. The authors should provide some biological insights.
3. Some case studies can be provided. For example, how the proposed method can be used to facilitate biological findings.

---

### Official Review · Reviewer_jV9G · 2024-11-03

**Soundness:** 3
**Presentation:** 3
**Contribution:** 3
**Rating:** 6
**Confidence:** 4

**Summary:**

The paper introduces GeneBench, a benchmarking suite specifically designed for evaluating Genomic Foundation Models across a wide range of genomic tasks. GeneBench includes evaluations of eleven GFMs on forty-four datasets, with tasks spanning various genomic regions and functions. This systematic benchmarking reveals insights based on the performance of GFMs across short- and long-range tasks. Furthermore, the paper proposes a new model that incorporates advantages from two types of models and demonstrates effective performance across all tasks.

**Strengths:**

- The paper offers a wide-range and detailed evaluation of GFMs and also provides concrete guidance for users on how to select models based on different tasks.
- The paper provides clear classification of GFMs and benchmarking tasks.
- Beyond benchmarking, the paper proposes a new model GeneBench based on the insights from the experiments, and achieves the best performance on most tasks.

**Weaknesses:**

- The paper belongs to a benchmarking paper, but it does not include comparisons with other existing genomic benchmarks (e.g., the length of input sequences, types of benchmarked methods, etc.). This limits the motivation why the research area needs this new benchmarking.
- While the benchmark focuses on GFMs, it's better to have simpler baselines without pretraining, (e.g. CNNs). Including such models would provide a deeper understanding of the advantages or limitations of GFMs relative to classical models.
- The paper doesn't provide sufficient descriptions on several tasks (e.g. Genomic Structure Prediction)

**Questions:**

- Could the authors discuss how GeneBench differs from or improves upon existing genomics benchmarking?
- Could the authors provide detailed input-output descriptions for some tasks (e.g. Genomic Structure Prediction)?
- For visualization in Figure 8, it would be helpful if the authors add x-axis and y-axis labels to the heatmaps. Similarly, for Figure 10 and 11, what's the range of y-axis?

---

### Official Review · Reviewer_HAWR · 2024-11-04

**Soundness:** 2
**Presentation:** 1
**Contribution:** 2
**Rating:** 3
**Confidence:** 4

**Summary:**

This paper introduces a benchmark framework for evaluating genomic foundation models. The authors gathered a large number of tasks from multiple existing papers for benchmarking. The tasks are classified into either long-range tasks or short-range tasks. A study that compares several existing genomic foundation models using the gathered tasks was performed. In addition, the authors proposed a hybrid approach that is supposed to work well for both short-range tasks and long-range tasks.

**Strengths:**

- Benchmarked large number of tasks.

- Compared all major genomic foundation models developed recently.

**Weaknesses:**

- The training in Pre-training is not clear. Should the optimization in Eq. (1) be actually involved in pre-training? The two categories of targets described in fine-tuning do not appear in Eq. (1) at all.

- The manuscript appears to be prepared in a hurry, needing major cleaning up. For example, there is no mentioning what the abbreviations used in Figure 4 stand for and in the caption of the same figure, it is mentioned (a), (b), and (c), but based on the content I believe only (c) is there.  As another example, in line 406, the authors mentioned they studied NT with different choices of number of parameters. However, there is no description of the results and respective discussion to offer any insights. As the last example, in Table 7, Caduceus is the second performing model, but the author said it was HyenaDNA in the text.

- There is no description on how the proposed Genhybrid was trained.

- The value of their main findings may be limited. Since attention-based models were trained using short-length sequences while convolution-based models trained using long sequences (to consider long context), it is expected to see the former is better in short-range tasks and the later has potential advantage in long-range tasks.

- The effort in data curation is minimal. It looks to me simply pulling from previous works.  Could the authors clarify if there is any additional processing or validation of the data.

- Due to the limitation of the work pointed by the authors themselves, I do not see they can answer the last two questions summarized in the second paragraph in their introduction. Could the authors explain?

**Questions:**

See weakness

---

### Official Review · Reviewer_MdoR · 2024-11-04

**Soundness:** 3
**Presentation:** 3
**Contribution:** 2
**Rating:** 5
**Confidence:** 4

**Summary:**

This paper introduced a benchmark suite for genomic foundation models(gFMs) called GeneBench that systematically evaluates gFMs on an wide array of datasets across a range of tasks for both short and long range sequence prediction. This work also presented a new method called GenHybrid that leverages both SSM and attention based model architectures.

**Strengths:**

1. Through and comprehensive dataset curation for gLM evaluation that covers both short and long range tasks.
2. A wide range on gFMs are benchmarked across various model architectures and parameter sizes.
3. The introduction of new hybrid method that leverages both attention based models and state-space models and outperformed existing models in most of the datasets evaluated.
4. In-depth analysis of the benchmarking results and provided insights into the current state of gFMs and their performance differential in various tasks.

**Weaknesses:**

1. Lacking on tasks beyond classification and regression. One of the most promising application of gFMs is to predict zero-shot mutation effects and generative modeling of genomic sequences. This benchmark effort misses in both aspects. Many mutation effect databases are available and a comprehensive curation of a benchmarking dataset will be of vast interest to the community.
2. Lacking vertical comparison of different model architectures on model sizes and pre-training schemes. I understand this will be computationally costly but as a benchmarking effort, this is necessary to paint a more complete picture of the model landscape.
3. Missing naive benchmarks and ab initio models for comparison. It has been shown in many recent studies that gFMs do not outperform ab initio models trained on task specific datasets. Adding both ab initio models and native benchmarks will be very  important as a benchmark suite.

**Questions:**

1. What’s the rationale behind the collection of tasks used in this study? they seem to be very similar in terms of task type and it will be more interesting to see more variation in tasks such as zero-shot mutational effect prediction and generative sequence modeling.
2. The paper is presented as a benchmark suite but the introduction of GenHybrid seems to be the main emphasis throughout the results section. However, the details of such model is missing from the main text of the paper. What’s the main focus of this paper?
3. Why not include ab initio models trained on tasks specific data and naive benchmark in this suite? The numers presented in the paper is without context can hardly can be used as a standardized benchmark for future methods.

---

### Public Comment · ~Yu_Bo1 · 2024-11-12

Fail to reproduce the results of Deepstarr using the provided scripts.

---

> ### Author Response · Authors · 2024-11-13
> **Thanks for your feedback**
>
> Dear friend,
>
> Thank you for your feedback. We’re sorry to hear about the difficulties reproducing DeepSTARR’s results. Could you provide more details regarding the specific issues or errors encountered during your experiments? This will help us identify potential discrepancies and support you more effectively. For reference, our experiments were conducted primarily on NVIDIA L40 GPUs, so any information on your setup and configurations might also be helpful in diagnosing the issue.

---

> > ### Public Comment · ~Yu_Bo1 · 2024-11-13
> >
> > Thank you for your response. To clarify, while I was able to reproduce the results in the current paper using the provided scripts, the result was different from the original DeepSTARR paper. Otherwise, I encountered no additional issues.

---

> > > ### Author Response · Authors · 2024-11-14
> > >
> > > Thanks for your attention. The main difference comes from the fact that we implemented DeepStar with pytorch, while the original version uses Tensorflow. The network structure and training strategy are copied from the original version. This benchmark is mainly to provide you with a reference of the performance of different models in a completely unified experimental environment, and we hope that it will be helpful to you.

---

### Note · Authors · 2025-01-20

I have read and agree with the venue's withdrawal policy on behalf of myself and my co-authors.